# Antibiotic prescribing and antimicrobial resistance: An evaluation of clinical teachers' knowledge, attitude and practices at a South African dental school

Suwayda Ahmed[1]*, Rukshana Ahmed[1], Razia Zulfikar Adam[1], Renier Coetzee[2]

1 Department of Prosthodontics, Faculty of Dentistry, University of the Western Cape, Cape Town, South Africa, 2 School of Public Health, University of the Western Cape, Cape Town, South Africa

* suahmed@uwc.ac.za

## Abstract

### Introduction

Indiscriminate antibiotic prescribing in healthcare continues to make a significant contribution to increasing global antimicrobial resistance rates. This public health threat has the potential to cause 10 million deaths per year, if no action is taken to address this phenomenon. In light of escalating AMR rates, the World Health Organization recommended a Global Action Plan against AMR in 2015, which highlighted global attention to AMR. One of the objectives of the Global Action Plan is to improve knowledge of antimicrobial resistance through effective education and training. Studies have suggested gaps in the knowledge and practice of antibiotic prescribing among healthcare professionals and undergraduate students, including medicine, dentistry, nursing, pharmacology and veterinary science. The dental profession has been identified as being a major contributor of excessive antibiotic prescribing, accounting for approximately 10% of systemic antibiotic prescriptions globally. Clinical knowledge, skills and practice gained during undergraduate dental programs can influence the clinical practice and competency of future prescribers. Clinical teachers thus play a critical role in shaping undergraduate dental students' prescribing behaviours. This translates to effective undergraduate healthcare programs which offer adequate education on antimicrobial resistance and rational antibiotic prescribing practices to ensure that students are well prepared prior to entering clinical practice.

### Aim

To determine the knowledge, attitudes and practices related to antimicrobial prescribing and AMR awareness among clinical teachers at a dental faculty in South Africa.

**Data availability statement:** All relevant data are within the manuscript and its Supporting information files.

**Funding:** The study was partially supported by the South African Medical Research Council Self-Initiated Research Fund. The funders had no role in the study design, data collection and analysis, decision to publish, or preparation of the manuscript.

**Competing interests:** The authors have declared that no competing interests exist.

**Abbreviations:** AMR: antimicrobial resistance; AMS: antimicrobial stewardship; GAP: Global Action Plan; WHO: World Health Organization.

## Methodology

A quantitative, cross-sectional questionnaire-based study was conducted among clinical teachers at the Faculty of Dentistry, University of the Western Cape in South Africa during the period of 28 September 2024 till 24 March 2025. A non-probability convenience sampling method was used. Responses were captured on a Google spreadsheet and exported to IBM SPSS Version 30 for descriptive and chi-square analyses.

## Results

Sixty-one clinical teachers participated in the study (75% female and 25% male), with an 87.1% response rate. Participants' work experience ranged from less than 5 years (2%), 5–10 years (11%), 11–15 years (18%), to more than 15 years (69%). The most commonly prescribed antibiotic regimen was a combination of amoxicillin and metronidazole. Majority of respondents (90%) prescribed a 5-day course of antibiotics, suggesting consistency with recommended short-course therapy. In cases of penicillin allergy, clindamycin (62%) was the most favoured alternative, which is concerning, as clindamycin is associated with numerous adverse effects. The understanding of AMR among clinical teachers was positive, with 92% correctly identifying AMR as bacteria resisting the effects of antibiotic therapy. Ninety-five percent of respondents viewed antibiotic resistance as a growing concern. The results demonstrate strong awareness and adherence to antibiotic prophylaxis guidelines among respondents, with 100% reporting both awareness of and compliance with such guidelines.

## Conclusion

The results reveal a range of knowledge, attitudes, and practices related to antibiotic prescribing and AMR awareness among clinical teachers, and indicate acceptable levels of knowledge and practice in various clinical scenarios, however certain identified prescribing practices may require targeted educational intervention. By strengthening antibiotic stewardship principles among clinical teachers, undergraduate prescribing practices can be further enhanced.

## Introduction

The increasing rate of antimicrobial resistance (AMR) is a global public health emergency and is widely referenced as a 'silent pandemic' [1,2]. Indiscriminate prescribing and misuse of antibiotics lead to increased AMR rates, thus contributing to this public health crisis [3]. It is estimated that 700000 deaths per annum are associated with AMR, with a predicted 10 million deaths per annum by 2050, if antibiotic use is not drastically reduced [4].

Research studies have identified the dental profession as a major contributor to improper antibiotic prescribing practices. Dentists are responsible for around 10%

of antibiotic prescriptions globally, and it is estimated that about 70% of these prescriptions are considered inappropriate [5–8]. This highlights the need to provide effective education on antibiotic prescribing and stewardship in undergraduate dental programs [9–11].

Clinical teachers play a crucial role in shaping undergraduate students' prescribing behaviours, as students often adopt the patterns and practices demonstrated during clinical training, with undergraduate students identifying clinical teachers as key influences in shaping their prescribing decisions. Consequently, the implementation of antimicrobial stewardship (AMS) principles into dental undergraduate training, specifically in relation to antibiotic prescribing, requires active engagement of clinical teachers and faculty. This participation is integral to rational prescribing knowledge, which needs to be consistently demonstrated and reinforced through undergraduate training [12,13]. Introducing education on AMR and AMS principles early in undergraduate programs is crucial to foster responsible prescribing behaviour once they enter clinical practice. Successful AMS requires the education of all healthcare professionals involved in antimicrobial handling, and this includes the medical, nursing, pharmacology, dental, and veterinary disciplines [14,15].

In response to the increasing rates of AMR, the World Health Organization (WHO) has developed various policies and frameworks to address this public health threat, including the Global Action Plan (GAP) on AMR, which requests member countries to adopt National Action Plans to address AMR. One of the objectives of the GAP on AMR is to improve knowledge of AMR through effective education and training, by proposing the integration of content and activities to strengthen AMR awareness and AMS principles as core components of health professions education [14,16]. The WHO GAP has emphasized the importance of training healthcare workers and undergraduate students on rational antibiotic prescribing and AMS, as an important way to prevent AMR [16]. In addition, the WHO has also drafted the WHO curricula guide for health workers' education and training on antimicrobial resistance. This guide provides an AMR competency framework, to encourage healthcare workers to make informed decisions based on best practices for antibiotic selection and prescribing, while also applying the principles of AMS [17].

The National Department of Health in South Africa has also been proactive, with the draft of its National AMR Strategy Framework [18]. This framework aligns with the WHO Global Action Plan as it highlights and encourages interdisciplinary efforts, antimicrobial stewardship and the increase of infection prevention and control measures [16]. The implementation of AMS at institutional levels is essential to this framework, with an emphasis on promoting education on AMR and stewardship principles through teaching and continuous training exercises [18]. Brink et al. (2017) have specifically highlighted the importance of increasing AMS efforts and expanding the undergraduate curriculum to address AMR among healthcare professionals, including those in dentistry, nursing and veterinary science within the South African healthcare system [19]. Prescribers of antibiotics have an important role in the effort to combat AMR. Inadequate education in undergraduate healthcare programs may contribute to the lack of confidence to correctly prescribe antibiotics. Thus, the importance of integrating comprehensive training on suitable antibiotic prescribing practices into curricula is essential, given the widespread use of antibiotics in clinical practice [20].

Accordingly, this study aimed to assess the self-perceived knowledge, attitude, and prescribing practices of clinical teachers at the Faculty of Dentistry, University of the Western Cape (UWC), to ensure that the information they impart is accurate and aligned with current evidence-based best practice. To the best of the authors' knowledge, this is the first study to examine the knowledge, attitudes and prescribing practices in relation to antibiotic prescribing and AMR among dental clinical teachers in South Africa.

## Methodology

### Study design

A quantitative cross-sectional analytical study design was used to assess the knowledge, attitude and prescribing practices of antibiotic prescribing and AMR awareness among clinical teachers at a dental school in South Africa

## Study setting

The study was conducted at the Faculty of Dentistry, University of the Western Cape (UWC), South Africa.

## Sample size

The sample size was calculated at a 95% confidence level and 5% precision for a finite population of N = 70, resulting in n = 60. All 70 eligible clinical teachers were invited to participate to maximise the response rate. A total of 61 responded resulting in an 87.1% response rate, which represents the majority of eligible study participants. The high response rate demonstrated a highly representative sample of the clinical teacher population at this faculty. As this is a single site study, using a convenience sampling method, the findings may not be generalizable to other dental schools in South Africa.

## Study duration

Data collection took place from 28 September 2024 till 24 March 2025.

## Sampling

A non-probability convenience sampling method was used, approaching the accessible participants at the Faculty of Dentistry, UWC; in order to maximize study participation.

## Inclusion criteria

Dentists who were employed as clinical teachers at the Faculty of Dentistry, University of the Western Cape, South Africa; with active involvement in clinical teaching or supervision during the study period.

## Exclusion criteria

Dentists who were not employed as clinical teachers at the Faculty of Dentistry, University of the Western Cape, South Africa, staff without clinical teaching responsibilities; and clinical teachers on extended leave during the data collection period were excluded from the study.

## Data collection

A questionnaire tool (see S1 Appendix), based on previously validated instruments and adapted to reflect the South African dental context was used in this study [21,22]. The questionnaires used by Loume et al. 2023 and Ealla et al. 2023 reflected the suitability of questionnaire items to evaluate the knowledge, attitude and practice among dental practitioners [21,22]. Questionnaires were distributed to the clinical teachers in a hard-copy format. The completed responses were collected from participants in sealed envelopes, and subsequently transferred and recorded on Google Forms by the primary investigator for data management and analysis. The Google Forms setting prevented multiple submissions and data entries were double checked to ensure accuracy. All data was password protected and with access restricted to the primary investigator. Participation was voluntary and anonymous with no identifying information collected. Informed consent was obtained from all participants prior to the questionnaire administration. An information sheet describing study information and objectives was distributed to participants prior to completion of the questionnaire. Clinical teachers were given 2 weeks to complete the questionnaire, with reminder messages sent weekly to increase the response rate.

A pilot study with a convenience sample (n = 10) was conducted two weeks prior to the main data collection period to validate the instrument through content and face validity. The questionnaire items were analysed using descriptive statistics, and no scales or composite scores were created, thus reliability testing was not performed. Experts in dentistry and antimicrobial resistance and stewardship reviewed the questionnaire for clarity and relevance. Face validity was established by the pilot participants who reflected the demographics of the study population and described the questions as

clear and appropriate. Following pilot study feedback, an 'Other, please specify' option was added to Question 3 to allow participants to indicate any guidelines they used, which was not specified in the questionnaire. Minor revision to the wording was made to improve understanding. The respondents of the pilot study were excluded from the main study.

The questionnaire collected demographic information, as well as participants' knowledge, attitudes, and practices regarding antibiotic prescribing and consisted of 26 multiple-choice and closed-ended questions. The questions captured the demographic information, including gender and years of clinical experience, and assessed participants' knowledge of antibiotic prescribing (e.g., alternative antibiotics for patients with penicillin allergies; and which antibiotic they most frequently prescribed). It also explored participants' attitudes toward antibiotic prescribing, including guideline use and factors that influence their prescribing habits. It also focused on participants' self-reported prescribing practices pertaining to antibiotic prescribing, such as frequency of antibiotic prescription and the clinical situations in which antibiotics are prescribed.

### Ethics statement

Ethical approval for this study was obtained from the Biomedical Research Committee at the University of the Western Cape (BMREC Reference Number: BM24/4/6). Written informed consent was obtained from respondents prior to the start of data collection and witnessed by the supervisor.

### Statistical analysis

The responses from the questionnaires were compiled in a Google Sheet for efficient data aggregation. The dataset was cleaned, coded, and exported to IBM SPSS (Statistical Package for the Social Sciences) Version 30.0 for further statistical analysis. Missing data was minimal for most items, with any missing responses excluded from the final analysis, and the percentages calculated on the valid responses captured. Descriptive statistics, including frequencies and percentages, were calculated for categorical variables such as gender, years of clinical experience, educational level, and prescribing-related variables. Associations between work experience and other categorical variables were assessed using the Chi-square test of independence. When expected cell counts were less than five, Fisher's Exact Test was applied to ensure statistical validity. A $p$-value less than 0.05 was considered statistically significant for all analyses. The results of inferential statistical analyses are reported in the Results section together with corresponding p-values.

## Results

### Demographic characteristics of participants

Participants were predominantly female (75%), with males representing 25% of the sample. Fifty-one percent were full-time clinical teachers, and the remaining 49% balanced both part-time teaching with private practice. Clinical experience varied, with 2% having less than 5 years, 11% having between 5–10 years, 18% having between 11–15 years, and 69% with more than 15 years of experience, as seen in Fig 1. Response rates varied by question. The percentages represent the proportion of valid responses for each item.

### Clinical teachers' knowledge of antibiotic prescribing and AMR

The results of this study, as seen in Table 1, presented a statistically significant association between years of clinical experience and the choice of alternative antibiotic prescribed in cases of penicillin allergy. In contrast, no statistically significant associations were observed between years of clinical experience and either the preferred antibiotic regimen or the duration of therapy. In cases of penicillin allergy, clindamycin was the most favoured alternative antibiotic (62%), followed by erythromycin (30%). A statistically significant association was found between years of clinical experience and antibiotic choice in penicillin-allergic patients (p = 0.025), indicating that clinical experience influenced antibiotic selection.

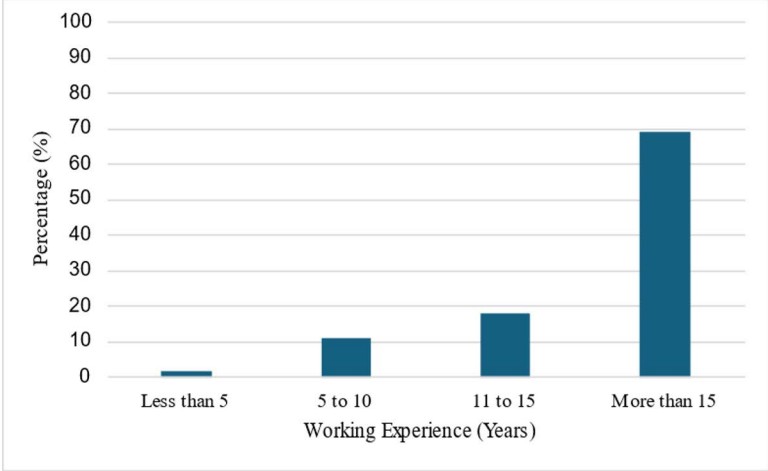

**Fig 1. Clinical teachers' clinical years of experience.**

**Table 1. Knowledge of antibiotic prescribing choice and duration of treatment.**

**Working Experience (Years)**

| | 0–10 | 11–15 | >15 | P-value |
|---|---|---|---|---|
| Which antibiotic do you prescribe most often for an adult patient with no medical allergies? | | | | |
| Amoxicillin | 2 (25.0%) | 2 (18.2%) | 13 (31.7%) | 0.306 |
| Amoxicillin and Clavulanic Acid | 2 (25.0%) | 1 (9.1%) | 11 (26.8%) | |
| Amoxicillin and Metronidazole | 4 (50.0%) | 7 (63.6%) | 17 (41.5%) | |
| Depends on what I am trying to treat | 0 (0.0%) | 1 (9.1%) | 0 (0.0%) | |
| If a patient is allergic to Penicillin, which antibiotic do you usually prescribe? | | | | |
| Azithromycin | 3 (37.5%) | 0 (0.0%) | 2 (4.9%) | 0.025 |
| Clindamycin | 4 (50.0%) | 8 (72.7%) | 25 (61.0%) | |
| Erythromycin | 1 (12.5%) | 3 (27.3%) | 14 (34.1%) | |
| What is the duration that you prescribe a course of antibiotics for? | | | | |
| 3 days | 0 (0.0%) | 0 (0.0%) | 1 (2.4%) | 0.543 |
| 5 days | 8 (100%) | 11 (100%) | 35 (85.4%) | |
| 7 days | 0 (0.0%) | 0 (0.0%) | 5 (12.2%) | |

Clinical teachers with more than 15 years of experience (61%) and those with 11–15 years of experience (72.7%) were more likely to prescribe clindamycin compared with those with less than 10 years of experience (50%). The most commonly prescribed antibiotic regimen among participants was a combination of amoxicillin and metronidazole (48%), followed by amoxicillin (28%) and amoxicillin–clavulanic acid (23%). However, no statistically significant association was observed between years of clinical experience and preferred antibiotic regimen (p=0.306), despite higher use of amoxicillin and metronidazole among the 11–15-year experience group. Similarly, no statistically significant association was found between years of clinical experience and the duration of antibiotic therapy (p=0.543). Overall, the majority of respondents (90%) prescribed a 5-day course of antibiotics, although 12.2% of clinical teachers with more than 15 years of experience reported prescribing a 7-day course.

Table 2 explores AMR awareness and knowledge, where 91.8% of respondents correctly described AMR as bacteria resisting the effects of antibiotics to which they were previously sensitive. Ninety-five percent of respondents believed

**Table 2. AMR knowledge and awareness.**

| Working Experience (Years) | | | | |
|---|---|---|---|---|
| | 0–10 | 11–15 | >15 | P-value |
| Do you think the frequent use of antibiotics might reduce the pharmacological treatment efficacy when using the same antibiotic again in the same patient? | | | | |
| Yes | 7 (87.5%) | 11 (100%) | 38 (92.7%) | 0.745 |
| No | 1 (12.5%) | 0 (0.0%) | 1 (2.4%) | |
| Unsure | 0 (0.0%) | 0 (0.0%) | 1 (2.4%) | |
| Do you think that antibiotic resistance is due to antibiotic prescription? | | | | |
| Yes | 5 (62.5%) | 7 (63.6%) | 30 (73.2%) | 0.405 |
| No | 3 (37.5%) | 2 (18.2%) | 5 (12.2%) | |
| Unsure | 0 (0.0%) | 2 (18.2%) | 6 (14.6%) | |
| Do you believe that antibiotic resistance is of growing concern? | | | | |
| Yes | 8 (100%) | 10 (90.9%) | 40 (97.6%) | 0.291 |
| No | 0 (0.0%) | 0 (0.0%) | 1 (2.4%) | |
| Unsure | 0 (0.0%) | 1 (9.1%) | 0 (0.0%) | |
| What is your primary source of updated information? | | | | |
| Scientifically published literature | 3 (37.5%) | 5 (45.5%) | 24 (58.5%) | 0.466 |
| Continuing dental education and conferences | 4 (50.0%) | 5 (45.5%) | 27 (65.9%) | 0.389 |
| Textbooks | 1 (12.5%) | 0 (0.0%) | 1 (2.4%) | 0.277 |
| Internet | 1 (12.5%) | 0 (0.0%) | 5 (12.2%) | 0.473 |
| Social media | 1 (12.5%) | 1 (9.0%) | 2 (4.9%) | 0.687 |
| Do you know about the AWaRe classification by WHO (World Health Organization) to use antibiotics? | | | | |
| Yes | 1 (12.5%) | 2 (18.2%) | 14 (34.1%) | 0.696 |
| No | 6 (75.0%) | 6 (54.5%) | 20 (48.8%) | |
| Unsure | 1 (12.5%) | 3 (27.3%) | 7 (17.1%) | |
| Do you know about the WHO Global Action Plan on AMR? | | | | |
| Yes | 1 (12.5%) | 1 (9.1%) | 13 (31.7%) | 0.201 |
| No | 6 (75.0%) | 5 (45.5%) | 16 (39.0%) | |
| Unsure | 1 (12.5%) | 5 (45.5%) | 12 (29.3%) | |

that frequent antibiotic use can reduce treatment efficacy, while 74% recognised that AMR is linked to prescribing practices. Although most respondents (95%) viewed AMR as a growing concern, only 28% were aware of the WHO AWaRe (Access, Watch, Reserve) classification and 25% knew about the WHO Global Action Plan on AMR [16,23]. Encouragingly, most clinical teachers rely on continuing education (59%) and scientific literature (54%) for information sources, though a small proportion reference social media (7%) or internet sources (10%). These results demonstrated that the participants years of clinical experience were not significantly associated with awareness of AMR, awareness of the WHO Global Action Plan, or knowledge of the AWaRe classification and the belief that antibiotic resistance is of growing concern. (p > 0.05).

## Clinical teachers' attitudes toward antibiotic prescribing

A strong awareness and adherence to antibiotic prophylaxis guidelines was observed among participants, with all participants reporting both awareness of and compliance with such guidelines as seen in Table 3. With respect to following antibiotic prescribing guidelines, the American Dental Association (38%) was the most frequently preferred, followed by

**Table 3. Antibiotic prophylaxis guidelines awareness and guideline adherence.**

|  | Frequency | Percentage |
|---|---|---|
| Aware of the guidelines for antibiotic prophylaxis | 61 | 100% |
| Follow guidelines for antibiotic prophylaxis | 61 | 100% |
| Which guidelines are you currently following? |  |  |
| • American Dental Association | 23 | 38% |
| • American Heart Association | 13 | 21% |
| • NICE (National Institute for Health and Care Excellence UK) Guidelines | 12 | 20% |
| • American Association of Endodontists Guidelines | 3 | 5% |
| • Standard Treatment Guidelines and Essential Medicines List | 8 | 13% |
| • Not sure | 7 | 11% |

the American Heart Association (21%), and the NICE guidelines (20%). A smaller proportion of participants followed the Standard Treatment Guidelines and Essential Medicines List [24] (13%), and 11% were unsure of the guideline source.

The results showed no statistically significant association between participants years of clinical experience and the clinical or non-clinical factors influencing antibiotic prescribing decisions, with all p-values > 0.05, as seen in Table 4. Across all clinical experience groups, prescribing decisions were most commonly based on patient symptoms, clinical protocols, and clinical experience. The decision to prescribe antibiotics mostly relies on patient symptoms (87%), followed by clinical protocols (75%), and clinical experience (56%), as seen in Table 4. Non-clinical factors include consideration of international guidelines (58%), recommendations from expert colleagues (43%), and undergraduate education (40%). Notably, 23% of respondents indicated that their prescribing decisions were influenced by patient preferences, while 22% were guided by the availability of specific antibiotics. Forty-eight percent of participants felt that their undergraduate training adequately prepared them for effective antibiotic prescribing and antimicrobial stewardship, while 43% disagreed with this statement and 10% were unsure.

**Table 4. Antibiotic prescribing attitude.**

**Working Experience (Years)**

|  | 0–10 | 11–15 | >15 | P-value |
|---|---|---|---|---|
| When prescribing antibiotics, do you base your decision on: |  |  |  |  |
| Patient symptoms | 6 (75.0%) | 11 (100%) | 36 (87.8%) | 0.241 |
| Clinical protocols | 4 (50.0%) | 10 (90.9%) | 31 (75.6%) | 0.125 |
| Clinical experience | 3 (37.5%) | 8 (72.7%) | 23 (54.1%) | 0.308 |
| What are the non-clinical factors that influence your antibiotic prescription? |  |  |  |  |
| Patient's preference | 1 (12.5%) | 4 (36.4%) | 9 (22.0%) | 0.446 |
| Autoimmune diseases | 0 (0.0%) | 0 (0.0%) | 3 (7.3%) | 0.481 |
| Knowledge obtained during undergraduate course | 4 (50.0%) | 5 (45.5%) | 15 (36.6%) | 0.716 |
| Availability at pharmacy | 4 (50.0%) | 2 (18.2%) | 7 (17.1%) | 0.112 |
| Recommendation from expert colleagues | 3 (37.5%) | 4 (36.4%) | 19 (46.3%) | 0.787 |
| International guidelines | 2 (25.0%) | 7 (63.6%) | 26 (63.4%) | 0.121 |
| Unavailable appointment for several weeks | 0 (0.0%) | 1 (9.1%) | 6 (14.6%) | 0.478 |
| Do you believe that your undergraduate education adequately prepared you to effectively prescribe antibiotics, and practice antimicrobial stewardship? |  |  |  |  |
| Yes | 5 (62.5%) | 3 (27.3%) | 20 (48.8%) | 0.496 |
| No | 2 (25.0%) | 6 (54.5%) | 18 (43.9%) |  |
| Unsure | 1 (12.5%) | 2 (18.2%) | 3 (7.3%) |  |

## Clinical teachers' antibiotic prescribing practices

Table 5 provides insight into antibiotic prescribing practices among clinical teachers. Although 58% indicated that the prescribing of antibiotics is not routine, antibiotics were still frequently prescribed for specific clinical situations. The most common indications include abscess of dental origin with fever (92%), increased risk of infective endocarditis (90%), and facial swelling (82%). Fewer respondents prescribed antibiotics for a localized dentoalveolar abscess (67%), and only 47% of respondents did so when a draining fistula was present. The prescribing of antibiotics solely for pain relief was uncommon (7%). When studying medical conditions that warrant antibiotic prophylaxis, most respondents indicated that infective endocarditis (92%) and rheumatic heart disease (90%) required prophylaxis, followed by heart defects (58%)

**Table 5. Antibiotic prescribing practices among clinical teachers.**

| Working Experience (Years) | | | | |
|---|---|---|---|---|
| | 0–10 | 11–15 | >15 | P-value |
| Is it common to prescribe antibiotics in your daily dental practice? | | | | |
| Yes | 4 (50.0%) | 4 (36.4%) | 14 (34.1%) | 0.002 |
| No | 4 (50.0%) | 3 (27.3%) | 27 (65.9%) | |
| Unsure | 0 (0.0%) | 3 (27.3%) | 0 (0.0%) | |
| For which of the following clinical situations do you prescribe antibiotics for | | | | |
| Abscess of dental origin with fever | 8 (100%) | 11 (100%) | 36 (87.8%) | 0.283 |
| Facial Swelling | 6 (75.0%) | 10 (90.9%) | 33 (80.5%) | 0.637 |
| Increased risk of infective endocarditis | 8 (100%) | 11 (100%) | 35 (85.4%) | 0.213 |
| Localized dentoalveolar abscess | 6 (75.0%) | 8 (72.7%) | 26 (63.4%) | 0.731 |
| Localized dentoalveolar abscess with draining fistula | 5 (62.5%) | 6 (54.5%) | 17 (41.5%) | 0.466 |
| Pain relief | 0 (0.0%) | 0 (0.0%) | 4 (9.8%) | 0.370 |
| Which medical condition/s do you consider prescribing antibiotic prophylaxis to, in order to avoid infective endo-carditis? | | | | |
| Diabetes mellitus | 0 (0.0%) | 0 (0.0%) | 4 (9.8%) | 0.370 |
| Autoimmune diseases | 0 (0.0%) | 0 (0.0%) | 3 (7.3%) | 0.481 |
| Immunosuppressive therapy | 1 (12.5%) | 1 (9.1%) | 4 (9.8%) | 0.966 |
| HIV/AIDS | 0 (0.0%) | 0 (0.0%) | 2 (4.9%) | 0.619 |
| Rheumatic heart disease | 6 (75.0%) | 10 (90.9%) | 38 (92.7%) | 0.311 |
| Heart bypass surgery | 2 (25.0%) | 3 (27.3%) | 10 (24.4%) | 0.981 |
| Pacemaker | 1 (12.5%) | 0 (0.0%) | 7 (17.1%) | 0.334 |
| Infective Endocarditis | 8 (100%) | 10 (90.9%) | 37 (90.2%) | 0.656 |
| Heart defects | 7 (87.5%) | 7 (63.6%) | 21 (51.2%) | 0.151 |
| Joint Prostheses | 5 (62.5%) | 5 (45.5%) | 19 (46.3%) | 0.689 |
| If antibiotic prophylaxis is indicated, what dosage do you prescribe? | | | | |
| Amoxicillin 1G - 1 hour before and 6 hours after | 0 (0.0%) | 0 (0.0%) | 1 (2.4%) | 0.722 |
| Amoxicillin 1G - 1 hour before treatment | 0 (0.0%) | 1 (9.1%) | 7 (17.1%) | |
| Amoxicillin 2G - 1 hour before treatment | 8 (100%) | 10 (90.9%) | 31 (75.6%) | |
| How many patients have you prescribed antibiotics to in the last two weeks? | | | | |
| 1-4 | 2 (25.0%) | 3 (27.3%) | 17 (41.5%) | 0.334 |
| 5-7 | 3 (37.5%) | 2 (18.2%) | 2 (4.9%) | |
| >7 | 0 (0.0%) | 0 (0.0%) | 1 (2.4%) | |
| None | 3 (37.5%) | 6 (54.5%) | 21 (51.2%) | |

and heart bypass surgery (25%). Conditions like HIV/AIDS (3%) and autoimmune diseases (5%) were rarely selected. On examination of the antibiotic prophylaxis dosage prescribed, 85% of participants prescribed 2g of Amoxicillin one hour before treatment, consistent with international recommended protocols. No statistically significant differences were found across experience groups for specific clinical conditions (p > 0.05), such as dental abscesses with fever, facial swelling, draining fistula and pain relief. Antibiotic prophylaxis, including dosage for certain medical conditions also demonstrated statistically significant association across experience groups. A statistical association was observed between the years of clinical experience and antibiotic prescribing frequency (p = 0.002), where 50% of the below 10-year group commonly prescribed antibiotics in their practice when compared to the 11–15-year (36.4%) and above 15-year (34.1%) experience group.

## Discussion

This study examined the knowledge, attitudes, and practices regarding antibiotics among clinical teachers at a dental school in South Africa—the first such study in this context. The findings demonstrated both strengths and significant gaps, with clinical teachers exhibiting appropriate antibiotic prescribing for specific clinical scenarios and guideline adherence. However, some knowledge gaps of WHO-initiated policies and frameworks, such as the WHO AWaRe classification (28%) and the WHO Global Action Plan (25%) were observed. Notably, most antibiotic prescribing behaviour remained constant across all years of experience. Statistical significance was observed with two variables among clinical teachers, namely, antibiotic preference in penicillin allergic patients (p = 0.025) and antibiotic prescribing frequency (p = 0.002).

Although antibiotic prescribing is an integral aspect of dental practice, varying results are observed in the literature relating to the selection of antibiotics for specific clinical conditions [25–27]. This emphasizes the importance of adhering to evidence-based prescribing guidelines and enhancing the education of clinical teachers, to ensure that accurate educational information is shared with students. The current study demonstrated the prescribing of antibiotics for various clinical conditions such as facial swelling, dental abscesses, increased risk of infective endocarditis and dental pain. No statistical association was observed between participants' year of experience and antibiotic prescribing for the various clinical situations. However, antibiotic prescribing frequency among participants demonstrated a statistical association (p = 0.002), where clinical teachers with less than 10 years' experience commonly prescribed antibiotics in practice (50%), compared to those with 11–15-years (36.4%) and above 15-years (34.1%) experience. Eriksen et al. (2025) conducted a similar study among dentists in Albania, which included academic staff, and found that antibiotic prescribing was common in everyday dental practice, with antibiotics being prescribed for both preventive and therapeutic needs. Eriksen et al. (2025) noted that 80.93% of respondents prescribed antibiotics for a dental abscess, 48.43% for pericoronitis and 16.86% for dental extractions, which is comparable to the results of the current study [28].

Similarly, Ramadan et al. 2019 studied prescribing practices among dentists at a teaching hospital in Sudan and found that both junior and senior staff routinely prescribed antibiotics for dental abscesses, pericoronitis, dry socket and post extraction, which differs with the findings of the current study. This practice also does not align with recommendations in various guidelines such as American Dental Association (ADA), National Institute for Health and Care and Excellence UK (NICE) guidelines and the WHO AWaRe classification [23,29–31].

The higher prescribing rate observed among the less experienced clinical teachers in the current study may indicate cautious clinical management in certain clinical situations, in comparison to more experienced clinical teachers, who may prefer direct operative management and clinical judgement to manage dental infections. The variations observed in the various studies, including the current study suggest that antibiotic prescribing practices may be influenced by institutional factors, clinical training environments and guideline availability; and not only on dentists' clinical experience. The most commonly prescribed alternative to penicillin in this study was clindamycin (62%), followed by erythromycin (30%) and azithromycin (8%). A statistical significance was observed (p = 0.025), with more experienced clinicians favouring clindamycin, which does not align with current antibiotic stewardship principles. These findings are similar to a previous study by

Farkas et al. (2021) among dentists in Croatia, which found the use of clindamycin as an alternative in the case of penicillin allergic patients, was prescribed by 86.6% of participants. In contrast, a study by Kandhro et al. (2024) in Hyderabad, clindamycin made up 10.02% of all prescriptions, suggesting greater adherence to recommended AMS principles [26,32].

It is important to note that the use of clindamycin as an alternative for patients with penicillin allergies has come under scrutiny due to its high risk of inducing Clostridioides difficile (C. difficile) colitis, its association with higher mortality compared to amoxicillin, and an increased risk of adverse drug reactions [33,34]. Consequently, international public health and professional regulatory bodies as well as various antimicrobial stewardship programs recommend limited use of clindamycin [33,35]. Studies have identified antibiotic prescribing by dentists being associated with community-associated C. difficile infections, with a single dose of clindamycin carrying a risk of causing a C. difficile infection [36–38]. A study investigating adverse drug reactions during antibiotic prophylaxis found that clindamycin was associated with higher rates of adverse drug reactions (most predominantly due to C. difficile infections), compared to amoxicillin; where 13 fatal and 49 non-fatal reactions per million prescriptions were linked to clindamycin. Amoxicillin was linked to 0 fatal reactions and 22.62 non-fatal reactions per 3 million prescriptions [38].

Various healthcare advisory bodies and guidelines, such as ADA Guidelines, NICE Antimicrobial Guidelines, and WHO AWaRe Classification have advocated for alternatives to clindamycin in the case of patients with penicillin allergies, which include cefuroxime, azithromycin and clarithromycin [23,30,31]. The current findings, together with evidence from the literature, suggest a potential knowledge practice gap in relation to clindamycin prescribing among clinical teachers that can be managed by targeted AMS interventions.

Limited knowledge was observed in the current study with respect to the awareness of WHO initiated stewardship policies, where only 25% of participants were aware of the WHO Global Action Plan on AMR and 28% knew of the WHO AWaRe classification. Similarly, in the study by Khalifa et al (2025) among dentists in Tunisia, only 9.3% of participants were aware of the WHO AWaRe classification [39]. WHO frameworks and policies are not necessarily focussed in undergraduate teaching and this may account for the limited knowledge evident in the current study. These initiatives and frameworks that are offered by the WHO inform antimicrobial stewardship programs, and may provide guidance and stimulus to educational programs; with the end goal of promoting public health and supporting health educators. Limited knowledge and awareness of the WHO Global Action Plan and the South African National Framework on AMR, while not compulsory, is useful to assess the alignment of international and national frameworks, which highlight AMS education as a tool to address increasing AMR rates.

The awareness of AMR among clinical teachers in this study is positive, as 95% of respondents view AMR as a growing concern. These findings are consistent with levels of AMR awareness among dental practitioners, as reported by D'Ambrosio et al., (2022), who reported AMR awareness at 98.9% [40]. The results also demonstrated that respondents are aware of and make use of antibiotic prescribing guidelines, although a range of guidelines are in use. These included international guidelines such as; the ADA guidelines, NICE guidelines, American Association of Endodontists (AAE) guidelines and the Standard Treatment Guidelines and Essential Medicines List for South Africa (STG-EML), with limited use of STG-EML observed [24,30,31,41].

The use of international guidelines may be due to greater familiarity by clinical teachers compared to the South African Standard Treatment Guidelines and Essential Medicines List with respect to dental prescribing, as the STG-EML has limited prescribing guidelines specific for dentistry [42]. These findings provide an opportunity to strengthen the use of STG-EML guidelines and highlights the need to incorporate dentistry-specific prescribing in these guidelines, as well as implementing continuing professional development programs for clinical teachers.

Undergraduate training around antibiotic prescribing plays a major role in prescribing practices among qualified dentists. Jones and Cope (2018) found that although participants felt they had received sufficient training, there was evidence of low confidence levels when deciding when to prescribe antibiotics and when to perform clinical operative treatment. The majority of their sample felt that additional undergraduate teaching on antibiotic prescribing and antimicrobial stewardship

would be beneficial [43]. In the current study, 48% percent of participants felt their undergraduate training adequately prepared them for effective antibiotic prescribing and antimicrobial stewardship, while 43% disagreed with this statement and 10% were unsure. These results suggest that the undergraduate antibiotic prescribing education of clinical teachers may not translate into confident clinical decision-making. Targeted educational intervention in relation to antibiotic prescribing and the integration of antimicrobial stewardship principles could possibly address this issue.

The current study has some limitations. These include the convenience sampling method, as well as the specific demographic and locational profile, as the researchers surveyed clinical teachers at UWC in the Western Cape, South Africa only. The convenience sampling method presents a measure of selection bias and this may affect the generalizability of the findings to other dental schools in South Africa. Self-reported data also introduces social desirability bias, and there could be an overstatement of findings.

Given that the rise of AMR rates worldwide is of great concern, education around correct antibiotic prescribing and the practice of effective antimicrobial stewardship has been highlighted as a means to alleviate this problem. Effective undergraduate dental education provides a means to address this challenge by ensuring that future dental prescribers are equipped to correctly prescribe antibiotics when required. This involves ensuring that both the curriculum and clinical teachers are appropriately informed and trained.

## Conclusion

This study revealed varying levels of knowledge, attitudes, and practices regarding antibiotic prescribing and AMR among clinical teachers, with acceptable performance in many clinical scenarios. However, the findings also underscore critical areas for improvement, particularly in the awareness of global antimicrobial stewardship frameworks with only 28% of clinical teachers aware of the WHO AWaRe classification and 25% familiar with the WHO Global Action Plan on AMR. Additionally, the findings revealed a high preference for clindamycin as an alternative for patients with penicillin allergies, in spite of its association with a higher risk of adverse effects. Targeted educational interventions and formal training programs must be developed to address these knowledge and practice gaps.

Faculty-specific guidelines aligned with national and international best practices should be implemented and continuously updated. Moreover, the integration of structured continuing professional development on antimicrobial resistance and stewardship into the responsibilities of clinical teachers must be prioritised. By equipping clinical teachers with enhanced, evidence-based knowledge, the accuracy and consistency of clinical teaching will be strengthened, ultimately ensuring that future dental practitioners are better prepared to prescribe antibiotics responsibly and contribute meaningfully to the fight against AMR.

## Limitations

Respondents in this study were recruited using convenience sampling based on accessibility rather than random selection. While this approach was practical for maximizing participation within the study timeframe, it may have introduced selection bias, and the sample may not fully represent all clinical teachers at the institution. Although this study showed a high response rate (87.1%), potential non-responder bias cannot be ruled out. The 13% of clinical teachers who did not participate may have differed from respondents in their knowledge, attitudes, or prescribing practices, potentially influencing the generalisability of the findings.

Quantitative data gained from questionnaires have the potential for social desirability bias, due to self-reporting of knowledge, attitudes and practices. The respondents in this study were clinical teachers at a dental school and, as such, they may have provided responses that they perceived as professionally or socially acceptable instead of reflecting their true knowledge, attitudes and practices. There could thus be an overstatement of findings. As this study was limited to a single dental school in the Western Cape, the findings may not be generalizable to other institutions, as there may be differences in their undergraduate course, teaching philosophy, resources, and other contextual factors, which may influence findings.

## Recommendations

Targeted continuing professional development, including workshops and lectures, should be implemented to reinforce evidence-based antibiotic prescribing practices and appropriate alternatives for penicillin-allergic patients among clinical teachers. Incorporating recognised antimicrobial stewardship frameworks and prescribing guidelines into both the curriculum and clinical teacher training programs may further support consistent and judicious prescribing behaviours. Additionally, the development of faculty-specific antibiotic prescribing guidelines would help standardise clinical decision-making and promote rational antibiotic use across all teaching departments.

## Supporting information

**S1 Appendix. Clinical teachers questionnaire.**
(DOCX)

## Author contributions

**Conceptualization:** Suwayda Ahmed, Razia Zulfikar Adam, Renier Coetzee.

**Data curation:** Suwayda Ahmed.

**Methodology:** Suwayda Ahmed, Rukshana Ahmed, Razia Zulfikar Adam, Renier Coetzee.

**Supervision:** Razia Zulfikar Adam, Renier Coetzee.

**Validation:** Rukshana Ahmed.

**Writing – original draft:** Suwayda Ahmed.

**Writing – review & editing:** Suwayda Ahmed, Rukshana Ahmed, Razia Zulfikar Adam, Renier Coetzee.

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
