## [Decision Letter · Decision Letter 0]

9 Dec 2025

PLOS One

Dear Dr. Ahmed,

Thank you for submitting your manuscript to PLOS ONE. After careful consideration, we feel that it has merit but does not fully meet PLOS ONE’s publication criteria as it currently stands. Therefore, we invite you to submit a revised version of the manuscript that addresses the points raised during the review process.

Dear Authors,

Please carefully read all the comments provided by the reviewers and address them accordingly, making the necessary changes in the revised manuscript.

Best regards and keep well

We look forward to receiving your revised manuscript.

Kind regards,

Mohmed Isaqali Karobari, BDS, MScD.Endo, Ph.D. Endo, FDS, FPFA, FICD, MFDS

Academic Editor

PLOS One

Journal Requirements:

South African Medical Research Council Self-Initiated Research Fun

5. Please ensure that you refer to Figure 1 in your text as, if accepted, production will need this reference to link the reader to the figure.

Additional Editor Comments:

Dear Authors,

Please carefully read all the comments provided by the reviewers and address them accordingly, making the necessary changes in the revised manuscript.

Best regards and keep well

Reviewers' comments:

Reviewer's Responses to Questions

**Comments to the Author**

1. Is the manuscript technically sound, and do the data support the conclusions?

Reviewer #1: Partly

Reviewer #2: Yes

Reviewer #3: Yes

2. Has the statistical analysis been performed appropriately and rigorously?

Reviewer #1: No

Reviewer #2: Yes

Reviewer #3: I Don't Know

3. Have the authors made all data underlying the findings in their manuscript fully available?

Reviewer #1: No

Reviewer #2: Yes

Reviewer #3: No

4. Is the manuscript presented in an intelligible fashion and written in standard English?

Reviewer #1: Yes

Reviewer #2: Yes

Reviewer #3: No

Reviewer #1: Overall Assessment

This study provides valuable insight into the knowledge, attitudes, and practices (KAP) related to antibiotic prescribing and AMR among dental clinical teachers in South Africa. The topic is highly relevant, aligns with global antimicrobial stewardship priorities, and fits well within PLOS ONE’s scope. The manuscript is generally well-structured and presents clear results.

However, several methodological, analytical, and interpretational issues require clarification before the manuscript can be considered for publication.

Major Comments

1. Methodology Requires Further Clarification

The sampling method is clearly described, but justification for using convenience sampling should be added, including its limitations.

Provide more details on questionnaire validation, including:

Specific changes made after the pilot study

Reliability statistics, if available

Please clarify if the questionnaire was adapted from validated tools or fully developed by the authors.

2. Statistical Analysis Section Needs Strengthening

Although descriptive statistics and chi-square tests are mentioned, no statistical test results (p-values, test statistics, effect sizes) are presented in the Results.

The authors must:

Add p-values for associations examined.

Indicate which variables showed statistically significant relationships.

Provide a brief interpretation of each.

3. Results Section Should Be More Analytical

Currently, results are descriptive. To enhance scientific value:

Include comparative analysis (e.g., years of experience vs. prescribing behaviour).

Present findings from chi-square analyses.

If no statistically significant relationships were found, state this explicitly.

4. Discussion Requires Deeper Critical Interpretation

The discussion is informative but:

It should integrate statistical associations (once added).

The issue of clindamycin overuse must be expanded with discussion of:

Adverse outcomes

International stewardship recommendations

Consider adding regional comparisons (Africa) in addition to global literature.

5. Limitations Section Needs Expansion

Include:

Sampling bias from convenience sampling

Self-report bias

Single-institution limitations

Potential non-responder bias

6. Language and Consistency

The manuscript is well written but requires:

Minor grammatical refinement

Consistency in terminology (e.g., “antibiotic prescribing patterns” vs “prescribing practices”)

Minor Comments

Ensure all acronyms are spelled out at first mention.

Add citations for WHO AWaRe, national guideline documents, and recent AMS literature (optional but recommended).

Figure 1 should include axis labels and clearer formatting.

Ensure tables follow PLOS ONE formatting guidelines.

Reviewer #2: The manuscript addresses an important and interesting topic aligning with global public health priorities.

1-Although the manuscript includes a Limitations section, these aspects should be more acknowledged in the Discussion, particularly: the single-institution sample limits generalizability, convenience sampling, and self-reported prescribing practices.

2-Please provide more discussion on the high dependence on clindamycin as the penicillin alternative. And whether this aligns with the emerging guidelines for antibiotics ?

3-Certain results are reported in both tables and text, leading to unnecessary repetition.

4-Minor typographical inconsistencies are present.

Reviewer #3: Thanks for this interesting paper. The power of this paper is its focus on the clinical tutors who influence undergraduate dentists when taking their first steps with patient management. This message is included in your manuscript but should be shining through even more strongly than at present - including in the abstract.

And this is a general feature of your manuscript - everything seems to be included but it's not just easy for the audience to find the highlights. My major suggestion is to use the CROSS reporting guideline for survey studies.

Abstract - misses some really important information - as per the point above - and in the methods section (e.g. when was this survey undertaken and in which dental school) - and doesn't make efficient use of word count e.g. by repeating points between the results (last line) and conclusion (first line)

Introduction - is too long so looses the power of its key points

Methods & results - ensure you cover all the points from the CROSS guideline.

Discussion - as per comments on the introduction. Much of the first three paragraphs would fit better in the abstract and/or introduction. In the third paragraph it's unclear why so much emphasis is placed on awareness of the WHO Global Action Plan. Clinical teachers certainly need to be aware of how it is has been translated into their context, but the need for awareness of the policy document seems curious. I have not gone through the discussion in detail because it really needs a significant redraft. When doing that, please remember to provide the reader with more information about key papers - for example in which countries were the studies by Sutej and by Hirayama undertaken? And in the next paragraph - it leaves the reader hanging. You need to provide more detail about how the cited studies vary rather than just citing 3 studies.

Good luck with the redraft. This will be an important addition to the literature, but it's not ready for publication yet.

**Do you want your identity to be public for this peer review?** For information about this choice, including consent withdrawal, please see our Privacy Policy

Reviewer #1: **Yes:** Bassam Abdul Rasool Hassan

Reviewer #2: No

Reviewer #3: No

---

## [Author Response · Author response to Decision Letter 1]

25 Jan 2026

Dear Editor and Reviewers

Thank you for all the feedback received. All changed have been highlighted in the manuscript

and explanation has been provided in the rebuttal letter.

Thank you

FEEDBACK ACTION

Reviewer 1

1. Methodology Requires Further Clarification.

The sampling method is clearly described, but justification for using convenience sampling should be added, including its limitations.

Provide more details on questionnaire validation, including:

Specific changes made after the pilot study

Reliability statistics, if available

Please clarify if the questionnaire was adapted from validated tools or fully developed by the authors. Acknowledged.

The methods section has been edited and all recommended feedback has been addressed.

Lines 169-172

Lines 186-194

Lines 196-197

Lines 201-203

Lines 206-209

2. Statistical Analysis Section Needs Strengthening

Although descriptive statistics and chi-square tests are mentioned, no statistical test results (p-values, test statistics, effect sizes) are presented in the Results.

The authors must:

Add p-values for associations examined.

Indicate which variables showed statistically significant relationships.

Provide a brief interpretation of each. Acknowledged.

This has been revised in the results section. Please see highlighted text in results section.

3. Results Section Should Be More Analytical

Currently, results are descriptive. To enhance scientific value:

Include comparative analysis (e.g., years of experience vs. prescribing behaviour).

Present findings from chi-square analyses.

If no statistically significant relationships were found, state this explicitly. Acknowledged.

The results section has been reworked to address the recommended feedback. All changes are highlighted in blue.

Lines 237-238

Lines 241-338

4. Discussion Requires Deeper Critical Interpretation

The discussion is informative but:

It should integrate statistical associations (once added).

The issue of clindamycin overuse must be expanded with discussion of:

Adverse outcomes

International stewardship recommendations

Consider adding regional comparisons (Africa) in addition to global literature.

Acknowledged.

The discussion has been substantially reworked.

Regional comparisons have been included.

Lines 340-458

5. Limitations Section Needs Expansion

Include:

Sampling bias from convenience sampling

Self-report bias

Single-institution limitations

Potential non-responder bias

Acknowledged.

The Limitations section has been rewritten to take all the recommended points into consideration.

Lines 478-484

Lines 489-492

6. Language and Consistency

The manuscript is well written but requires:

Minor grammatical refinement

Consistency in terminology (e.g., “antibiotic prescribing patterns” vs “prescribing practices”)

Ensure all acronyms are spelled out at first mention.

Add citations for WHO AWaRe, national guideline documents, and recent AMS literature (optional but recommended).

Figure 1 should include axis labels and clearer formatting.

Ensure tables follow PLOS ONE formatting guidelines. Acknowledged.

Prescribing practices used throughout the manuscript

Acronyms spelled out

Manuscript has been edited for grammatical errors.

Citations provided for WHO AWaRe, South African national guideline documents (References 16, 23 and 24)

Lines 229 and 296

Reviewer 2

Although the manuscript includes a Limitations section, these aspects should be more acknowledged in the Discussion, particularly: the single-institution sample limits generalizability, convenience sampling, and self-reported prescribing practices. Acknowledged.

The limitation section has been rewritten.

Lines 478-492

Please provide more discussion on the high dependence on clindamycin as the penicillin alternative. And whether this aligns with the emerging guidelines for antibiotics? Acknowledged.

More detail provided in the discussion and has been rewritten.

Lines 340-458

Certain results are reported in both tables and text, leading to unnecessary repetition. Acknowledged.

The results section has been edited and reworked.

Lines 241-338

4-Minor typographical inconsistencies are present. Acknowledged.

Manuscript has been edited for grammatical errors.

Reviewer 3

Abstract - misses some really important information - as per the point above - and in the methods section (e.g. when was this survey undertaken and in which dental school) - and doesn't make efficient use of word count e.g. by repeating points between the results (last line) and conclusion (first line) Acknowledged.

The abstract has been rewritten to address reviewers’ feedback.

Lines 67-72

Lines 78-79

Lines 98-100

Introduction - is too long so looses the power of its key points Acknowledged

The introduction has been reorganised and unnecessary detail has been removed.

Methods & results - ensure you cover all the points from the CROSS guideline.

The methods section has been edited to address feedback and follow CROSS guidelines

The result section has been edited and certain areas have been rewritten.

Lines 291-338

Discussion - as per comments on the introduction. Much of the first three paragraphs would fit better in the abstract and/or introduction. In the third paragraph it's unclear why so much emphasis is placed on awareness of the WHO Global Action Plan. Clinical teachers certainly need to be aware of how it is has been translated into their context, but the need for awareness of the policy document seems curious. I have not gone through the discussion in detail because it really needs a significant redraft. When doing that, please remember to provide the reader with more information about key papers - for example in which countries were the studies by Sutej and by Hirayama undertaken? And in the next paragraph - it leaves the reader hanging. You need to provide more detail about how the cited studies vary rather than just citing 3 studies. Acknowledged

Paragraphs have been moved to the introduction.

More detail and explanation have been provided for cited studies.

Explanation for WHO Global Action Plan, South African National AMR Framework has been provided in discussion.

The discussion has been reworked.

Lines 340-458

---

## [Decision Letter · Decision Letter 1]

10 Feb 2026

Dear Dr. Ahmed,

Thank you for submitting your manuscript to PLOS ONE. After careful consideration, we feel that it has merit but does not fully meet PLOS ONE’s publication criteria as it currently stands. Therefore, we invite you to submit a revised version of the manuscript that addresses the points raised during the review process.

We look forward to receiving your revised manuscript.

Kind regards,

Mohmed Isaqali Karobari, BDS, MScD.Endo, Ph.D. Endo, FDS, FPFA, FICD, MFDS

Academic Editor

PLOS One

**Journal Requirements:**

**Additional Editor Comments:**

Dear Authors,

Please carefully read all the reviewers' comments and address them accordingly, making the necessary changes in the revised manuscript.

Best regards and keep well

Reviewers' comments:

Reviewer's Responses to Questions

**Comments to the Author**

Reviewer #1: (No Response)

Reviewer #2: All comments have been addressed

2. Is the manuscript technically sound, and do the data support the conclusions?

Reviewer #1: (No Response)

Reviewer #2: Yes

3. Has the statistical analysis been performed appropriately and rigorously?

Reviewer #1: (No Response)

Reviewer #2: Yes

4. Have the authors made all data underlying the findings in their manuscript fully available?

Reviewer #1: (No Response)

Reviewer #2: Yes

5. Is the manuscript presented in an intelligible fashion and written in standard English?

Reviewer #1: (No Response)

Reviewer #2: Yes

Reviewer #1: (No Response)

Reviewer #2: • The revised manuscript has been improved.

• Please ensure that the "perceived knowledge" vs. "objective knowledge" distinction is clear in the introduction, as the study relies on self-reported data

• Please provide consistency in terminology between "antibiotic prescribing patterns" and "prescribing practices".

**Do you want your identity to be public for this peer review?** For information about this choice, including consent withdrawal, please see our Privacy Policy

Reviewer #1: **Yes:** Bassam Abdul Rasool Hassan

Reviewer #2: No

---

## [Author Response · Author response to Decision Letter 2]

17 Feb 2026

The Editor PLOS One

16 February 2026

Dear Editor

Re: Resubmission of manuscript to PLOS One Journal PONE-D-25-62403R1

I am pleased to submit the revision of our manuscript entitled ‘Antibiotic Prescribing and Antimicrobial Resistance: An Evaluation of Clinical Teachers' Knowledge, Attitude and Practices at a South African Dental School’ for your consideration for publication in PLOS One Journal, following feedback received.

Based on feedback received please find the rebuttal with regards to points in the manuscript.

FEEDBACK ACTION

Please ensure that the "perceived knowledge" vs. "objective knowledge" distinction is clear in the introduction, as the study relies on self-reported data Acknowledged and completed

The following paragraph is in the introduction when describing the current study: Accordingly, this study aimed to assess the self-perceived knowledge, attitude, and prescribing practices of clinical teachers at the Faculty of Dentistry, University of the Western Cape (UWC), to ensure that the information they impart is accurate and aligned with current evidence-based best practice.

Please provide consistency in terminology between "antibiotic prescribing patterns" and "prescribing practices" Acknowledged and completed

The entire manuscript has been edited to only use ‘antibiotic prescribing practices’

If the reviewer comments include a recommendation to cite specific previously published works, please review and evaluate these publications to determine whether they are relevant and should be cited. There is no requirement to cite these works unless the editor has indicated otherwise. Acknowledged

No references were added to the manuscript on the request of the reviewers.

Please review your reference list to ensure that it is complete and correct. If you have cited papers that have been retracted, please include the rationale for doing so in the manuscript text, or remove these references and replace them with relevant current references. Any changes to the reference list should be mentioned in the rebuttal letter that accompanies your revised manuscript. If you need to cite a retracted article, indicate the article’s retracted status in the References list and also include a citation and full reference for the retraction notice. Acknowledged and references have been checked for correctness.

A detailed explanation for the addition and removal of references will be provided in a separate table below.

Previous feedback received recommended that the introduction and discussion was unnecessarily long with unnecessary detail and needed to be reworked.

The introduction and discussion sections of the manuscript has been substantially reworked and this required the removal and addition of various references.

I apologize for any confusion arising from changes to the reference list between the original submission and the resubmission, which were due to compilation and editing errors. The references have now been corrected, and additional references have been included where necessary to fully address the reviewers’ feedback. Further explanation for the referencing is provided below.

Herewith follows the explanation for references:

In the introduction the following reference was added:

World Health Organization (2019) Health workers’ education and training in antimicrobial resistance on Antimicrobial Resistance Curricula Guide. https://iris.who.int/bitstream/handle/10665/329380/9789241516358-eng.pdf which is the correct reference for the WHO AMR Curricula Guide.

The following references were removed from the introduction:

Reasons: Based in reviewer feedback- ‘The Introduction is too long and loses the power of its key points’ and also the need to strengthen focus on ‘clinical tutors who influence undergraduate dentists’ we removed the following during our reworking of the introduction:

These references were originally included to provide broader contextual background; however, following the reviewer’s feedback, the associated text was identified as unnecessary and redundant. In an effort to streamline the introduction and improve clarity and readability for the reader, this section and the accompanying references have been removed.

1. Yang C, Xie J, Chen Q, Yuan Q, Shang J, Wu H, et al. Knowledge, Attitude, and Practice About Antibiotic Use and Antimicrobial Resistance Among Nursing Students in China: A Cross Sectional Study. Infection and Drug Resistance. 2024;Volume 17:1085–1098. doi:10.2147/idr.s454489.

2. Kafle K, Blondel-Hill E, Essack S, Grayson L, Levy Hara G, Nathwani D, Shetty N. Health workers' education and training on antimicrobial resistance: curricula guide. Geneva, Switzerland: World Health Organization; 2019. Available from: https://apps.who.int/iris/bitstream/handle/10665/329380/9789241516358-eng.pd

3. Nukaly H, Aljuhani R, Alhartani M, Alhindi Y, Asif U, Alshanberi A, et al. Knowledge of Antibiotic Use and Resistance Among Medical Students in Saudi Arabia. Advances in Medical Education and Practice. 2024;Volume 15:501–512. doi:10.2147/amep.s462490.

4. Alshehri AA, Khawagi WY. Knowledge, Awareness, and Perceptions Towards Antibiotic Use, Resistance, and Antimicrobial Stewardship Among Final-Year Medical and Pharmacy Students in Saudi Arabia. Antibiotics. 2025;14(2):116. doi:10.3390/antibiotics14020116.

5. Bonna AS, Mazumder S, Manna RM, Pavel SR, Nahin S, Ahmad I, et al. Knowledge attitude and practice of antibiotic use among medical students in Bangladesh: A cross-sectional study. Health Science Reports. 2024;7(9). doi:10.1002/hsr2.70030.

6. Bajalan A, Bui T, Salvadori G, Marques D, Schumacher A, R¨osing CK, et al. Awareness regarding antimicrobial resistance and confidence to prescribe antibiotics in dentistry: a cross-continental student survey. Antimicrobial Resistance Infection Control. 2022;11(1). doi:10.1186/s13756-022-01192-x.

7. Ahmed S, Ahmed R, Adam RZ, Coetzee R. Antimicrobial resistance, antibiotic prescribing practices and antimicrobial stewardship in South Africa: a scoping review. JAC-Antimicrobial Resistance. 2024;7(1). doi:10.1093/jacamr/dlaf014.

8. Chigome A, Ramdas N, Skosana P, Cook A, Schellack N, Campbell S, et al. A Narrative Review of Antibiotic Prescribing Practices in Primary Care Settings in South Africa and Potential Ways Forward to Reduce Antimicrobial Resistance. Antibiotics. 2023;12(10). Available from: https://www.mdpi.com/2079-6382/12/10/1540. doi:10.3390/antibiotics12101540.

9. Aljeldah MM. Antimicrobial Resistance and Its Spread Is a Global Threat. Antibiotics. 2022;11(8):1082. doi:10.3390/antibiotics11081082.

The following references were added in the discussion

1. World Health Organisation. (2022). The WHO AWaRe (Access, Watch, Reserve) antibiotic book. https://www.who.int/publications/i/item/WHO-MHP-HPS-EML-2022.02

2. Ramadan AM, Al Rikaby OA, Abu-Hammad OA, Dar-Odeh NS. Knowledge and Attitudes Towards Antibiotic Prescribing Among Dentists in Sudan. Pesqui bras odontopediatria clín integr. 2019;19: 1–10. doi:10.4034/PBOCI.2019.191.17

3. Lockhart PB, Tampi MP, Abt E, Aminoshariae A, Durkin MJ, Fouad AF, et al. Evidence-based clinical practice guideline on antibiotic use for the urgent management of pulpal- and periapical-related dental pain and intraoral swelling. The Journal of the American Dental Association. 2019;150: 906-921.e12. doi:10.1016/j.adaj.2019.08.020 PMID: 31668170

4. Richards D. Prophylaxis against infective endocarditis. Evid Based Dent. 2008;9: 37–38. doi:10.1038/sj.ebd.6400577 PMID: 18583999

5. Thornhill MH, Dayer MJ, Durkin MJ, Lockhart PB, Baddour LM. Risk of Adverse Reactions to Oral Antibiotics Prescribed by Dentists. J Dent Res. 2019;98: 1081–1087. doi:10.1177/0022034519863645 PMID: 31314998

6. Chitnis AS, Holzbauer SM, Belflower RM, Winston LG, Bamberg WM, Lyons C, et al. Epidemiology of Community-Associated Clostridium difficile Infection, 2009 Through 2011. JAMA Intern Med. 2013;173: 1359. doi:10.1001/jamainternmed.2013.7056 PMID: 23780507

7. Bye M, Whitten T, Holzbauer S. Antibiotic Prescribing for Dental Procedures in Community-Associated Clostridium difficile cases, Minnesota, 2009–2015. Open Forum Infectious Diseases. 2017;4: S1–S1. doi:10.1093/ofid/ofx162.001

8. Thornhill M, Dayer M, Prendergast B, Baddour L, Jones S, Lockhart P. Incidence and nature of adverse reactions to antibiotics used as endocarditis prophylaxis. J Antimicrob Chemother. 2015;70:2382–8.

9. Ben Hadj Khalifa A, Boukhris H, Ayari G, Ben Mansour DE, Baccouche C. Knowledge, attitudes and behaviors of Tunisian dentists on the prescription of antibiotics. Cumhuriyet Dental Journal. 2025;28: 361–369. doi:10.7126/cumudj.1656296

10. American Association of Endodontists. AAE guidance on antibiotic prophylaxis for patients at risk of systemic disease. American Association of Endodontists; 2017 Available from: https://www.aae.org/specialty/wpcontent/uploads/sites/2/2017/06/aae_antibiotic-prophylaxis.pdf

11. Govender T, Suleman F, Perumal-Pillay VA. Evaluating the implementation of the standard treatment guidelines (STGs) and essential medicines list (EML) at a public South African tertiary institution and its associated primary health care (PHC) facilities. J of Pharm Policy and Pract. 2021;14: 105. doi:10.1186/s40545-021-00390-z PMID: 34906236

12. Jones E, Cope A. Knowledge and attitudes of recently qualified dentists working in Wales towards antimicrobial prescribing and resistance. Eur J Dental Education. 2018;22. doi:10.1111/eje.12387 PMID: 30125439

Previous reviewers’ comments Reasoning for addition of references

The discussion is informative but:

It should integrate statistical associations (once added).

The issue of clindamycin overuse must be expanded with discussion of:

Adverse outcomes

International stewardship recommendations.

Please provide more discussion on the high dependence on clindamycin as the penicillin alternative. And whether this aligns with the emerging guidelines for antibiotics? Clindamycin safety and adverse outcomes data was discussed, the following additional references were included:

Thornhill et al 2015., Chitnis et al. 2013, Bye et al.2017; Thornhill et al. 2019

Add citations for WHO AWaRe, national guideline documents, and recent AMS literature (optional but recommended). Recommendation for WHO AWaRe citation.

Addition of the following reference- The WHO AWaRe (Access, Watch, Reserve) Antibiotic Book. 1st ed. Geneva: World Health Organization; 2022.

Consider adding regional comparisons (Africa) in addition to global literature This has been addressed and the following references has been added:

Ben Hadj Khalifa et al. (2025)

Ramadan et al. (2019)

The power of this paper is its focus on the clinical tutors who influence undergraduate dentists when taking their first steps with patient management. This message is included in your manuscript but should be shining through even more strongly than at present Greater focus on clinical teachers and role in new prescribers/undergraduate training. The following reference was added:

Jones & Cope (2018)

And whether this aligns with the emerging guidelines for antibiotics? The following references were added to illustrated guideline recommendations and use:

Richards (2008)

Lockhart et al. (2019)

American Association of Endodontists (2017)

Govender et al. 2021

I have not gone through the discussion in detail because it really needs a significant redraft. When doing that, please remember to provide the reader with more information Based on this feedback. The introduction and discussion were significantly reworked, and required the removal of certain refences, as well as addition of references to address the comments of the 3 reviewers.

The following references were removed from the discussion:

In response to the reviewers’ feedback that the Discussion "needs a significant redraft" and studies were cited without providing significant detail, the following references were removed during the discussion restructure. This included citing studies which were relevant to the regional context as recommended. Studies were removed to eliminate overlap and provide a more focused discussion. The result is a more focused discussion with depth for greater understanding and ease of reading to the reader. The references that were removed mostly related to the text in manuscript that was unnecessary as indicated by reviewers feedback, therefore not all these references have specific replacements.

1. Karasneh RA, Al-Azzam SI, Ababneh M, Al-Azzeh O, Al-Batayneh OB, Muflih SM, et al. Prescribers’ Knowledge, Attitudes and Behaviors on Antibiotics, Antibiotic Use and Antibiotic Resistance in Jordan. Antibiotics. 2021;10(7):858. doi:10.3390/antibiotics10070858.

2. Šutej I, Bašić K, Šegović S, Peroš K. Antibiotic Prescribing Trends in Dentistry during Ten Years’ Period—Croatian National Study. Antibiotics. 2024;13(9):873. doi:10.3390/antibiotics13090873.

3. Hirayama K, Kanda N, Hashimoto H, Yoshimoto H, Goda K, Mitsutake N, et al. Antibiotic Prescription Trends in Dentistry: A Descriptive Study Using Japan’s National Database. Journal of Public Health Dentistry. 2025;85(2):153–159. doi:10.1111/jphd.12663.

4. Bajalan A, Bui T, Salvadori G, Marques D, Schumacher A, R¨osing CK, et al. Awareness regarding antimicrobial resistance and confidence to prescribe antibiotics in dentistry: a cross-continental student survey. Antimicrobial Resistance Infection Control. 2022;11(1). doi:10.1186/s13756-022-01192-x.

5. Fine P, Leung A, Bentall C, Louca C. The Impact of Confidence on Clinical Dental Practice. European Journal of Dental Education. 2018 12;23. doi:10.1111/eje.12415.

6. Thompson W, Mceachan R, Pavitt S, Douglas G, Bowman M, Boards J, et al. Clinician and Patient Factors Influencing Treatment Decisions: Ethnographic Study of Antibiotic Prescribing and Operative Procedures in Out-of-Hours and General Dental Practices. Antibiotics. 2020;9(9):575. doi:10.3390/antibiotics9090575.

7. Contaldo M, D’Ambrosio F, Ferraro GA, Di Stasio D, Di Palo MP, Serpico R, et al. Antibiotics in Dentistry: A Narrative Review of the Evidence beyond the Myth. International Journal of Environmental Research and Public Health. 2023;20(11):6025. doi:10.3390/ijerph20116025.

8. Rodríguez-Fernández A, Vázquez-Cancela O, Piñeiro-Lamas M, Herdeiro MT, Figueiras A, Zapata-Cachafeiro M. Magnitude and determinants of inappropriate prescribing of antibiotics in dentistry: a nation-wide study. Antimicrobial Resistance Infection Control. 2023;12(1). doi:10.1186/s13756-023-01225-z.

9. Kerr I, Reed D, Brennan AM, Eaton KA. An investigation into possible factors that may impact on the potential for inappropriate prescriptions of antibiotics: a survey of general dental practitioners’ approach to treating adults with acute dental pain. British Dental Journal. 2021. doi:10.1038/s41415-021-3008-x.

I hope that all concerns have been addressed, and a sufficient explanation has been provided .

Thank you for considering our manuscript. We look forward to the opportunity to contribute to your journal. Kind regards

---

## [Decision Letter · Decision Letter 2]

2 Mar 2026

Antibiotic Prescribing and Antimicrobial Resistance: An Evaluation of Clinical Teachers' Knowledge, Attitude and Practices at a South African Dental School

PONE-D-25-62403R2

Dear Dr. Ahmed,

We’re pleased to inform you that your manuscript has been judged scientifically suitable for publication and will be formally accepted for publication once it meets all outstanding technical requirements.

Kind regards,

Mohmed Isaqali Karobari, BDS, MScD.Endo, Ph.D. Endo, FDS, FPFA, FICD, MFDS

Academic Editor

PLOS One

Additional Editor Comments (optional):

Dear Authors,

The authors have addressed all the reviewers' comments and suggestions, and the manuscript has undergone significant improvement. I would like to congratulate the authors and wish them all the very best in their future endeavours.

Best regards and keep well.

Reviewers' comments:

Reviewer's Responses to Questions

**Comments to the Author**

Reviewer #1: (No Response)

Reviewer #2: All comments have been addressed

2. Is the manuscript technically sound, and do the data support the conclusions?

Reviewer #1: (No Response)

Reviewer #2: Yes

3. Has the statistical analysis been performed appropriately and rigorously?

Reviewer #1: (No Response)

Reviewer #2: Yes

4. Have the authors made all data underlying the findings in their manuscript fully available?

Reviewer #1: (No Response)

Reviewer #2: Yes

5. Is the manuscript presented in an intelligible fashion and written in standard English?

Reviewer #1: (No Response)

Reviewer #2: Yes

Reviewer #1: (No Response)

Reviewer #2: The authors have addressed all the comments and suggestions, which has significantly enhanced the clarity, depth, and overall quality of the revised manuscript.

**Do you want your identity to be public for this peer review?** For information about this choice, including consent withdrawal, please see our Privacy Policy

Reviewer #1: **Yes:** Bassam Abdul Rasool Hassan

Reviewer #2: No

---

## [Editor Report · Acceptance letter]

PONE-D-25-62403R2

PLOS One

Dear Dr. Ahmed,

I'm pleased to inform you that your manuscript has been deemed suitable for publication in PLOS One. Congratulations! Your manuscript is now being handed over to our production team.

Kind regards,

on behalf of

Prof Dr. Mohmed Isaqali Karobari

Academic Editor

PLOS One